# Self-Recovery Prompting: Promptable General Purpose Service Robot System with Foundation Models and Self-Recovery

**Abstract:** A general-purpose service robot (GPSR), which can execute diverse tasks in various environments, requires a system with high generalizability and adaptability to tasks and environments. In this paper, we first developed a top-level GPSR system for worldwide competition (RoboCup@Home 2023) based on multiple foundation models. This system is both generalizable to variations and adaptive by prompting each model. Then, by analyzing the performance of the developed system, we found three types of failure in more realistic GPSR application settings: *insufficient information*, *incorrect plan generation*, and *plan execution failure*. We then propose the *self-recovery prompting pipeline*, which explores the necessary information and modifies its prompts to recover from failure. We experimentally confirm that the system with the self-recovery mechanism can accomplish tasks by resolving various failure cases.
https://sites.google.com/view/srgpsr-anon

**Keywords:** Foundation Models, Service Robotics, Self-Recovery

## 1 Introduction

A general-purpose service robot (GPSR) is a concept aiming to develop a robot system that accomplishes various types of human requests likely to happen in real-world environments [1]. As the system needs to handle various types of requests in various environments, it has to be generalized between them. Besides, to enhance usability, the system is required to handle ambiguous commands in natural interaction with humans, such as speech, which might have insufficient information to understand properly without communication or leveraging common sense knowledge.

Recent progress in foundation models [2], a set of large pre-trained models with diverse datasets, has brought high generalization performances in perception and task planning from natural language to robotics. Furthermore, these models can be adapted to various tasks and environments with *prompting* [3], a technique to enhance the performance of the models by modifying the inputs without additional training. However, most of the robot learning studies utilize foundation models as modules, and there is a lack of discussions about the system design or integration and evaluation of complex environments such as household environments.

This paper first presents a robot system that won the GPSR task in RoboCup Japan Open (RCJ) 2023 and second place in RoboCup (RC) 2023. The GPSR task held in RoboCup aims to benchmark the performance of entire generalized robotic systems based on the concept of GPSR mentioned above. To avoid confusion, we use GPSR to represent a task itself and GPSR to represent a concept throughout the paper. The competitions are held in a household environment, and robots are required to perform various tasks asked by a human operator. Figure 1 shows an example of requests accompanied by the sequence of output of our system, which integrates multiple foundation models, including *GPT-4* [4] for planning, *Whisper* [5] for speech recognition, and *Detic* [6] and *CLIP* [7] for object recognition (Figure 2). In short, our system uses GPT-4 as the core of the system to generate the plan and the other three models to convert human requests and environmental information into text information or recognize part of the environment specified in the text. Notably, our system can

Submitted to the 7th Conference on Robot Learning (CoRL 2023). Do not distribute.

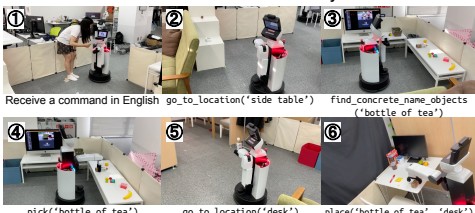

Figure 1: Example of GPSR task execution by our system. The given commands are converted into a sequence of skills that can be executed by the robot and then executed one by one.

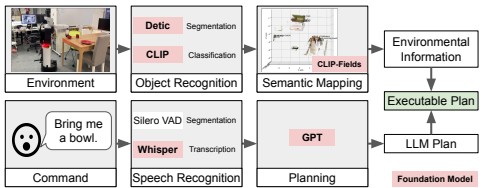

Figure 2: Overview of our foundation-model-based system. The foundation models collaborate to process the environment and a natural language command into an executable plan.

be entirely *promptable*, meaning we can easily tune the system only by specifying system prompts (without model training). In section 3, we describe more detailed integrations of each foundation model and provide evaluations at both the per-module and the whole system level.

While the achievement of our system in the GPSR task supports the importance of foundation models to realize the concept of general-purpose service robots from the point of generalization and adaptation, there are still several issues regarding its performance; the system still cannot perfectly execute complex requests due to the accumulation of errors in each module, and the entire system becomes difficult to tune as the system grows larger (or by using more foundation models). More critically, the current GPSR task abstracts some desiderata of the GPSR concept due to the nature of robot competition. For example, most of the information on objects (names, categories, and locations) is given before the task starts, and thus, there is no need to judge whether the information is sufficient or not. In section 4.1, we categorize three types of failure modes of the current robot system to achieve GPSR systems, namely 1) *insufficient information*, 2) *incorrect plan generation*, and 3) *plan execution failure*, and discuss the requirements of the robot system.

Based on the discussion, we then introduce a *self-recovery mechanism* on top of the above-mentioned GPSR system to further enhance the system's versatility. Here, self-recovery means a system that retries to accomplish the original requests somehow when the system encounters some failure. While the notion of self-recovery is simple and has been implemented in various robot systems, we tailored it for promptable robot systems (i.e., systems that can improve performance only by adding or modifying their prompt). Specifically, we design a pipeline, called *self-recovery prompting*, which refines their prompts by past experiences and active communication with the operator. For the experiments, we handcraft seven types of commands that require retry associated with the aforementioned three failure modes of the original system and show that our system can recover from various types of failures.

## 2 Preliminaries & Related Works

### 2.1 General Purpose Service Robot (GPSR)

The concept of GPSR is introduced in Walker et al. [1] wherein robots are expected to perform diverse tasks given by humans in a natural manner (e.g., verbal communication). According to the concept of GPSR, RoboCup@Home league [8] tests the performance of GPSR as GPSR task [9]. The GPSR task is held in a real-world household environment, and the robots are expected to perform tasks given verbally by the operator (referee) as perfectly as they can within a time limit. The tasks are generated randomly with the command generator [10]. Since the rule is updated every year, we adapt the rule for RoboCup 2023[1] in this paper.

### 2.2 Foundation Models for Robotics

Foundation models are a set of models trained on broad datasets at scale and adaptable to a wide range of downstream tasks [2], such as large language models (LLMs) [11, 4, 12, 13], vision-language models (VLMs) [7, 6, 14, 15, 16], and audio-language models (ALMs) [5, 17, 18]. A

---

[1] https://github.com/RoboCupAtHome/RuleBook/tree/706764626baf073d56ab2e61c1a3c5d3c339cfb4

key characteristic of the foundation models is their generalization and adaptation ability, thanks to pre-training on massive and diverse datasets (often collected from the Internet). Especially, several foundation models, including Whisper [5], GPT [11, 4], CLIP [7], and Detic [6], are *promptable*; they can enhance the performance by adding text description to the input (called *prompting*) about the contexts such as detailed instruction [19] and environmental information [20].

In the robotics community, foundation models are utilized as modules for perception and planning. As for the perception, VLMs such as CLIP [7], Detic [6], and SAM [15] are utilized in object and environmental perception [19]. Similarly, ALMs such as Whisper [5] and AudioCLIP [18] are used for speech [21] and sound [22] recognition. In addition, several robot systems use LLMs as task planners. LLMs are expected to handle the ambiguity of natural language and convert them to machine-interpretable representations, reasoning missing information in commands. For example, SayCan [23] utilizes LLMs to generate plans given from natural language instructions such as *"I spilled my drink, can you help?"*. As the variants, Code as Policies [24] generates Python codes (including calls of external perception modules) and executes them, and Obinata et al. [25] propose to generate state machine [26] using LLMs.

The closest setting and systems to ours is Obinata et al. [25], which proposes a solution for GPSR task using foundation models in recognition and planning. While the usage of LLMs for planning and VLMs for object detection is similar to ours, we further utilize foundation-model-based modules for speech recognition and semantic mapping and exceeded their performance in GPSR task in RoboCup@Home Japan Open 2023. In addition, we discuss the typical failure cases and introduce a novel self-recovery mechanism into the foundation-model-based robot system (section 4).

### 2.3 Robot System with Self-Recovery

In the context of robotics, the importance of the notion of self-recovery has been emphasized and implemented in the motion planning of multi-legged robots [27] and in the mechanical design of aerial robots [28]. This paper aims to realize self-recoverable task planning in GPSR systems under the framework of prompting with foundation models.

Some concurrent robot learning studies using foundation models provide solutions for managing failures in plan execution. For example, DoReMi [29] proposes to detect failures of skill execution via VLMs and replan if the skill fails. FindThis [30] proposes to resolve the ambiguation in object recognition through the dialogue between humans and robots. Ren et al. [31] presents a framework to ask humans for help in an interactive manner if the uncertainty of the appropriate plan is high. In contrast, this paper presents the entire GPSR system in real-world household environments, which is promptable and has functions to autonomously address multiple types of failures.

## 3 Promptable System for GPSR Task

In this section, we first introduce our promptable GPSR system with foundation models, which achieved second place in GPSR task and won third prize in RoboCup@Home 2023.

For the realization of a GPSR system, multiple foundation models with high generalization and adaptability were leveraged for the system in this study. The following five models (four of which are foundation models, and one is a model that consists of an integration of foundation models) have the ability to enhance the system to be generalized and adaptive with prompting: Whisper [5] for speech recognition, GPT-4 [4] for task planning, Detic [6] for object detection and segmentation, CLIP [7] for object classification, and CLIP-Fields [32] for integration of environmental information. Figure 2 shows an example of how the foundation models can be used in our proposed system.

For all the experiments, we used HSR (Human Support Robot) developed by Toyota Motor Corporation [33] in the real world. The experiments were conducted in a real-world simulated household environment with several rooms, such as a living room, a dining room, and a study room.

### 3.1 Overview

#### 3.1.1 Speech Recognition

Speech recognition consists of two modules: a voice activity detection (VAD) module and a transcription module. Silero VAD [34] is used for VAD, and Whisper [5] is used for transcription. Since

Whisper is promptable with natural language, transcription performance can be enhanced using prior knowledge about task settings, such as names of humans, objects, and locations.

### 3.1.2 Object Recognition

The object recognition module consists of an object detection module and an object classification module. Detic [6] and CLIP [7], both of which are promptable foundation models, were used for detecting and classifying detected objects, respectively. We leverage the feature that these models accept open-vocabulary text inputs as prompts for object detection and classification, while conventional pre-trained models usually have fixed classes. For object detection, we prompt information of objects of interest (e.g., object name, category, description) into Detic. Then, the images segmented by Detic are classified with CLIP based on similarities between the embeddings of text description of the target objects and the embeddings of the segmented images.

### 3.1.3 Planning

To convert a natural language command into an executable format, we leverage GPT-4 [4] in our system. We prepare 21 skill functions (Table 1) that can accomplish given commands if appropriately combined. The desired output is an array where skill functions, including their arguments, are correctly arranged in the order they are executed by the robot in JSON format [35].

The task planning process is based on the Chain-of-Thought prompting [36, 37] and has a two-step structure. The first step is dividing the command into minimal steps and deciding the order for the robot to perform in natural language. For example, the command *"bring me an apple from the dining table"* is converted into an array of sentences such as *"Move to the dining table," "Find apple,"* and similarly. The array continues in the order of execution. In the second step, skill functions to be used with their arguments (e.g., locations, object names) are decided for each sentence leveraging function calling of GPT-4. By providing examples of the commands and their desired responses as prompts, it is possible to specify the output format and improve task planning accuracy.

### 3.1.4 Semantic Mapping

We integrate environmental information into a 3D semantic map using CLIP-Fields [32], which utilize three foundation models: Detic for object recognition, CLIP for image encoding, and Sentence BERT [38] for image label encoding. The robot can refer to the environmental information in CLIP-Fields for task planning.

## 3.2 Experiments of Each Module

### 3.2.1 Speech Recognition

We first compared the speech recognition performance with and without prompts. The prompt includes object names, human names, and location names (i.e., room and furniture) that may appear in commands. 12 commands were used for the experiments. The commands were generated by the command generator used in the Enhanced General Purpose Service Robot (EGPSR) task of RoboCup@Home 2023. 14 people participated in this study. For each command, the examinees were asked to read it aloud once to reduce misread cases. Then, they were asked to read the same command twice, and their voices were recognized by the robot. The typical cases from the obtained results are indicated in Table 2. The use of location names in advance shows a reduced likelihood of variations in interpreting location names. This suggests that pre-defined location names as prompts are an effective technique for improving transcription performance.

### 3.2.2 Task Planning

The planning performance between using tuned prompts and minimal prompts was examined in comparison. To test the effect of providing a prompt on the LLM's reasoning ability of translation from given commands into the sequence of execution steps, we compared the result of the first step of the task planning (described in section 3.1.3.)

The tuned prompt was adjusted so that most of the generated commands from the command generator [10] used in the EGPSR task are correctly converted into arrays of sentences. This prompt consisted of the settings of the environment, the situation the robot was in, and the iteration of example commands and their ideal responses. Since it was impossible to align the LLM output (i.e.,

an array of the sentences) without any prompt, the minimal prompt (shown below) was designed with minimum sufficient content for eliciting the output format.

> *You are a helpful assistant for a robot. The robot is in a house. Your mission is to convert natural language command into a list of sentences. The robot will execute the sentences in order to complete the task.*

The commands used in this experiment were the same as in section 3.2.1. The success or failure of planning for each output was judged by whether the command was completed when the robot performed each skill function perfectly.

As a result, in many cases, the plan generated with the minimal prompt was inappropriate, while the plan with the tuned prompt was executable. Some commands and their outputs of each prompt are shown in Table 3. The outputs of the minimal prompt lacked necessary preliminary action or contained sentences that could not be related to any skill function. Therefore, it can be said that providing instructions as a prompt is effective in eliciting LLM to generate executable plans.

### 3.2.3 Object Recognition

Object recognition performance was evaluated in comparison between setting Detic for open-vocabulary mode with prompts, and closed-vocabulary mode without prompts. Experiments were conducted using images with the same member of objects throughout the experiment.

CLIP was used consistently with prompts, and for both open-vocabulary Detic and CLIP, prompts were tuned using images of the same objects placed in different locations and orientations. For instance, the prompts for "white rope" and "jump rope" were set as follows.

> Prompts of a white rope and a jump rope for Detic
> *"rope": "a photo of a tangled white rope",*
> *"jump rope": "a photo of a green jump rope, a type of toy"*

> Prompts of a white rope and a jump rope for CLIP
> *"white rope": "a photo of a white rope",*
> *"jump rope": "a photo of a green jump rope"*

Validation experiments were conducted using entirely new images. Every object detected by Detic was cropped by its bounding box and classified by CLIP.

Figure 3 shows that when Detic was used in open-vocabulary mode with the prompts shown above, it correctly detected the white rope, which was present in the closed-vocabulary case but remained undetected. During the segmentation phase with Detic, the white rope was misidentified as a green jump rope. Nevertheless, by incorporating prompts, even for objects with similar shapes, segmentation accuracy improved, and when applied to CLIP, correct recognition, as demonstrated in this case, could be expected. The result suggests the potential for improved recognition accuracy.

### 3.3 Results of RoboCup@Home `GPSR` task

We participated in RoboCup@Home DSPL (Domestic Standard Platform League) of RoboCup Japan Open (RCJ) 2022 and 2023 and RoboCup (RC) 2023 (worldwide). The proposed system was evaluated in RoboCup Japan Open 2023 and RoboCup 2023. In the competitions, the scores of the `GPSR` task were respectively given when speaking the transcribed command and accomplishing the task. It should be noted that the case where the robot autonomously requested human help and continued the command execution was also regarded as a success, with a reduction of scores afterward. Conspicuously, in our trial of RoboCup Japan Open 2023, all the commands were completed within the time limit. The team scored 170 points, the perfect score for the second to the most challenging category (Category 2). The team's place in the GPSR task and overall are indicated in Table 4. Figure 4 illustrates scores of `GPSR` task in RoboCup Japan Open (2022 and 2023). Our team marked more than 180 % of the second-placed team in 2023.

## 4 Self-Recovery Mechanism for Promptable Robot System

In the previous section, we proposed the entire system for `GPSR` task in RoboCup@Home, which can achieve top-level performance. However, owing to the nature of robot competition, some desiderata of GPSR are abstracted in `GPSR` task. For instance, the majority of information regarding objects (names, categories, and locations) is provided prior to the task, removing the necessity to assess whether the command contains sufficient information. Besides, since the time is limited, skipping

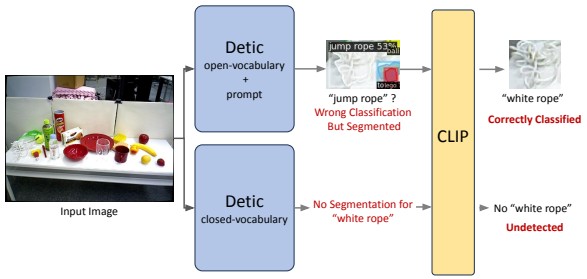

Figure 3: Image recognition results depending on open and closed vocabulary modes of Detic. With prompts added for open-vocabulary mode case, as shown in section 3.2.3, "white rope," undetected with closed-vocabulary mode, is successfully detected in the end with open-vocabulary mode.

Figure 4: GPSR score results in recent years. The figure shows that the system developed by us (indicated in red) marked more than 180% of the second-placed team in RCJ 2023.

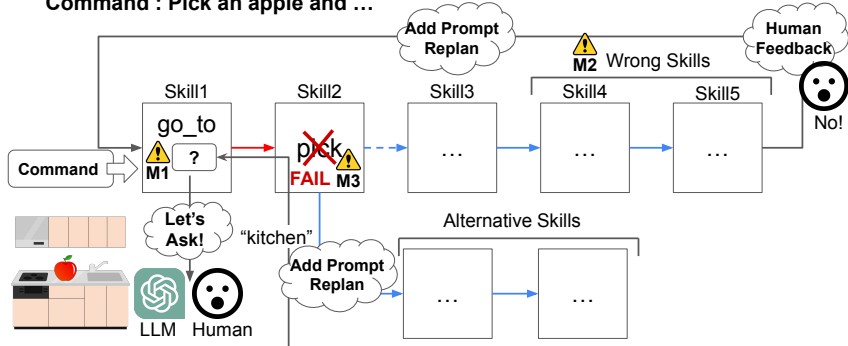

Figure 5: Example of three failure modes of GPSR systems and prompt-based self-recovery mechanisms. M1: Location information is lacking. The robotic system asks for a human or LLM and adds obtained information into their prompt. M2: On the realization of the wrong performance, the system re-plans. M3: On the realization of execution failure, the per-skill recovery function is activated.

the task has been a better approach for achieving higher scores instead of finding a recovery plan when the robots once failed to execute the commands. Therefore, achieving a higher score or winning in GPSR task is not sufficient to achieve genuine GPSR systems.

In this section, we first classify challenges for attaining authentic GPSR. Ideally, GPSR can be achieved with complete information about the environment, the ability to generate correct plans (skill sequences), and the perfect execution of the skills in each plan. However, in general, these three assumptions are often violated and challenges to realizing the authentic GPSR concept. Here we analyze issues that often occur in GPSR systems and organize the failure modes of GPSR systems into three patterns, namely, *insufficient information*, *incorrect plan generation*, and *plan execution failure*. Then, we propose to add a *self-recovery mechanism* into the system and evaluate the performance under the settings of the aforementioned three failure modes.

### 4.1 Three Failure Modes of GPSR Systems

**(M1) Insufficient Information**

In a domestic environment, robots have to perform in a dynamic environment; for example, the locations of objects and humans are ever-changing. Moreover, registering all the information about the environment (e.g., object or human names, categories, and locations) to the system beforehand is not feasible. Even if the system has enough reasoning or recognition ability of human intent, lacking information about the environment prohibits the system from generating the correct plan at once.

For example, the information necessary to plan can be lacking in many ways, such as *"I lost my watch. Could you find it for me?"* (a situation where even humans do not know the location of the objects), or *"Could you bring me a cup?"* (a situation that humans have assumed where it should be but not clarified in the command).

**(M2) Incorrect Plan Generation**

Even when the system has information sufficient to accomplish the task (i.e., no insufficient information problem), the current system in section 3 cannot perfectly accomplish the task. For example,

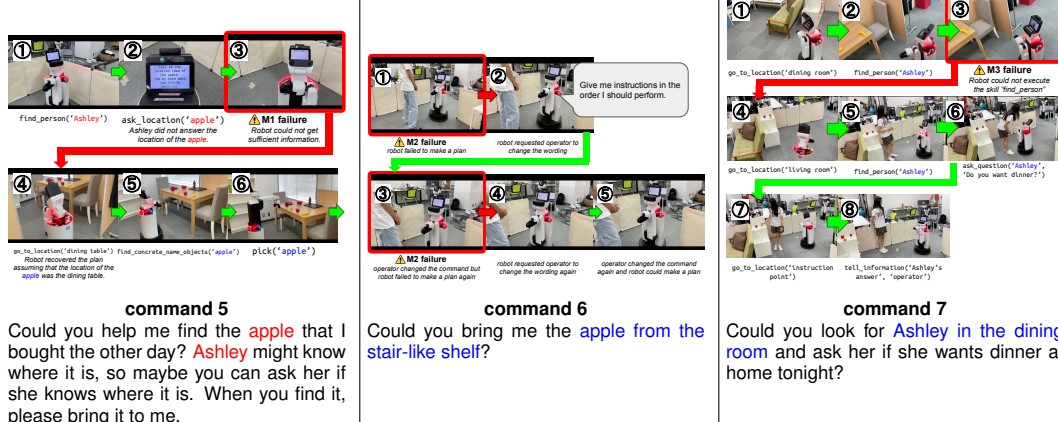

Figure 6: Example of three failure modes and execution of our system with self-recovery prompting. Commands 5, 6, and 7 in Table 5 correspond to left, middle, and right, respectively. The red box highlighted areas indicate failure patterns at each command. The green arrow indicates a normal plan transition, and the red arrow indicates that a recovery plan has been triggered.

the robot can catch noises along with spoken commands and mistranscribe them, which leads to the generation of a wrong plan. Moreover, wrong plans can be generated due to a lack of reasoning performance or common sense of the planner. Suppose a simple case where the planner is just to extract verbs in the order of appearance in the commands and make them into executable skill sequences, and the command is *"Could you fetch me an apple?"*, the plan may start with "Bring the apple to the operator," which is a mistake. Instead, the ideal plan is to "go to the location where the apple is (estimate if necessary before that)", "find it", "grab it", and then "bring it back to the operator".

**(M3) Plan Execution Failure**

Even if the plan is generated correctly with sufficient information on the environment, the robot system may fail to execute skills in the environment. This failure mode is due to the imperfection of the skill execution and is inevitable in the nature of real-world systems. For example, the robots may compute the wrong manipulation poses of objects and fail to grasp objects. The inability to find an object or false positives is also considered an execution failure. It is important to note that execution failure not only occurs because of such hardware execution errors but can also be attributed to the environment of service (i.e., the object does not exist in the house).

### 4.2 Self-Recovery Prompting Pipeline

In order to deal with the three failure modes for realizing GPSR systems, we introduce a self-recovery mechanism from the failure modes with prompting, called *self-recovery prompting pipeline*, as illustrated in Figure 5. In concrete, we developed a self-recoverable GPSR system as an entire system by adding functions of replanning and human-robot interaction based on the foundation-model-based system described in section 3.

#### 4.2.1 Recovery for Insufficient Information (M1)

In the case of insufficient information (M1), the missing information necessary for planning is supplemented with common sense that the planning module has (e.g., food is likely to be in the kitchen or dining room) and additional information obtained by talking with humans (e.g., asking where the apple is). In concrete, we implement two recovery functions into our GPSR system. For the case that the location name (e.g., dining table) is not included in the command (or dialogue with humans), the system first infers the candidates of location from the command leveraging an LLM-based planner and plans to visit them. In the case that an operator or the LLM output refers to a location name not defined in the robot system, the robot asks the operator to rephrase the location name and extract it using LLM from the operator's response.

#### 4.2.2 Recovery for Incorrect Plan Generation (M2)

In the case of incorrect plan generation (M2), we develop solutions for it regarding command recognition and plan generation. As for the command recognition, the promptable speech recognition

module (e.g., Whisper) can be improved by updating the prompts as described in section 3. For plan generation, the prompts for the LLM-based planner are updated reflecting human feedback given to the system after finishing the original plan to confirm task completion. If the task is not evaluated as completed, another plan is regenerated with the planner with updated prompts.

### 4.2.3 Recovery for Plan Execution Failure (M3)

In the case of plan execution failure (M3), the failure can be recovered per-skill and per-plan. For per-skill recovery, we develop two functions; one is to retry skill execution in the plan (e.g., retry navigation skill), and the other is to replan alternative skill sequence using the following prompt template instead of executing the original skill.

> *The robot is supposed to {*`task_content`*} . The robot tried to {*`failed_action`*} {*`robot_at`*}, but failed. What should the robot do next?*

Per-plan recovery is performed when the task is considered a failure in the human feedback after the execution of the entire plan, similar to the solution of the 2nd failure mode (M2). In this case, the prompts of the LLM-based planner are updated with the feedback, and the entire plan is regenerated and executed. For example, this occurs when a wrong object from the specified object is recognized in object recognition skill. After completing the plan, the system asks the operator to provide more information about the objects, especially the name and color. Prompts for the object recognition module are updated, and the task plan is regenerated.

## 4.3 Experiments

### 4.3.1 Experiment Setup

Experiments were conducted to examine whether the system can recover from each of the failure modes by leveraging the proposed system. The system is tested in a domestic environment similar to that of section 3. The difference from the setting in the previous section is that object and human names and their locations are not given in advance of the task (the map with the location names is given). Following the experimental purposes, the commands used for the tests are created manually instead of generated with the command generator, and all commands are expected to be too challenging to complete with the original system in section 3. Table 5 represents the prepared commands. The checkmark (✓) in the table indicates that the command and its setups have characteristics of the corresponding failure modes.

### 4.3.2 Results

For all tested commands, our self-recovery prompting mechanism successfully resolved failures. Three of the seven results, which represent examples of recovery functions in accordance with M1, M2, and M3 are explained in detail below and illustrated in Figure 6.

For the case of the 5th command, the robot asked Ashely for the location of the apple but received no response, thus potentially causing the system to stop due to lack of information. However, the developed system overcame this potential failure point by seeking general knowledge of LLM (ask the location of *"apple"*) in this phase. For the case of the 6th command, since the instruction contained a phrase that was difficult to transcribe (*"apple from the stair-like shelf"*), it was difficult for the robot to generate a plan. Our system overcame this failing point by requesting the operator rephrase the command. For the case of the 7th command, execution failure at the finding person phase was a possible failing point. The system recovered from it by re-planning.

## 5 Discussion and Conclusion

In this paper, we first developed promptable GPSR systems utilizing multiple foundation models, which can achieve top-level performance in the worldwide competition (RoboCup@Home 2023). By analyzing the performance of the developed system, we organized three failure modes in more realistic GPSR applications: *insufficient information*, *incorrect plan generation*, and *plan execution failure*. We then proposed the self-recovery prompting pipeline, which leverages the prompting of the system to overcome each failure mode, and evaluated the entire system using seven handcrafted commands. To enhance further studies in GPSR systems with self-recovery, benchmarks equipped with adaptive human-robot interaction will be essential to standardize the performance, which may also be realized with LLMs and VLMs.

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

# A Appendix

## A.1 Skill Functions Prepared in the System

Table 1 shows the 21 skill functions we have prepared for the system in this paper. See section 3.1.3 for detailed explanations.

Table 1: 21 Skill Functions.

| Functions | Arguments | Descriptions |
| --- | --- | --- |
| `go_to_location` | `location` | navigate the robot to {`location`} |
| `ask_location` | `object` | get location name of {object} by asking human using VAD and Whisper, if unsuccessful, by asking LLM |
| `find_concrete_name_objects` | `object` (opt:room) | find {`object`} using Detic and CLIP in the {`room`} |
| `find_category_name_objects` | `category` (opt:room) | find {`category`} objects using Detic and CLIP in the {`room`} |
| `count_concrete_name_objects` | `objects` | count the number of {`objects`} using Detic and CLIP |
| `count_category_name_objects` | `category` | count the number of {`category`} objects using Detic and CLIP |
| `find_person` | `person` | find {`person`} using Keypoint R-CNN [39] |
| `detect_person_pose` | `person` | detect {`person`} 's pose using Keypoint R-CNN |
| `find_specific_pose_person` | `person` `pose` | find {`person`} with {`pose`} using Keypoint R-CNN |
| `count_specific_pose_person` | `person` `pose` | count the number of {`person`} with {`pose`} using Keypoint R-CNN |
| `count_person` | | count the number of person using Keypoint R-CNN |
| `follow_person` | `person` (opt:location) | follow {`person`} to {`location`} using YOLOv8 [40] |
| `guide` | `person` `location` | guide {`person`} to {`location`} |
| `pick` | `object` `location` | pick {`object`} at {`location`} |
| `hand_over` | `object` `person` | hand over {`object`} to {`person`} |
| `ask_person_to_hand_over` | `object` `person` `query` | ask{`person`} to hand over {`object`} by saying {`query`} |
| `place` | `object` `location` | place {`object`} on {`location`} |
| `ask_question` | `person` `question` | say {`question`} to {`person`} and get answer using VAD, Whisper, and LLM |
| `answer_question` | (opt:person) | answer to {`person`}'s question using VAD, Whisper, and LLM |
| `tell_information` | `information` `person` | tell {`information`} to {`person`} using LLM |
| `operate_door` | `location` `operation` | {`operation`} (open/close) the door at {`location`} |

## A.2 Experiment Results of Each Module

Table 2 shows the results of the speech recognition module in our system comparing with and without prompts. See section 3.2.1 for the experiment conditions. Table 3 shows the results of the LLM-based task planner in our system, comparing the tuned prompts and minimal prompts. See section 3.2.2 for the experiment settings.

Table 2: Comparison of transcription results (without and with prompts) for speech recognition with Whisper.
**Command:** *Go after the person at the bed please*.

| w/o Prompts | w/ Prompts |
|---|---|
| person of the band, please | person at the bed please |
| person at the bat place | person at the bed please |

**Command:** *Offer something to drink to all the people dressed in white in the bedroom*.

| w/o Prompts | w/ Prompts |
|---|---|
| dressed in white in the bathroom | dressed in white in the bedroom |

Table 3: Comparison of generated plans (with minimal prompts and with tuned prompts) with GPT-4. "Success" indicates that a plan that would satisfy the command if each skill function was performed perfectly was generated.
**Command:** *Describe the objects on the kitchen table to me please*

| Minimal | Tuned |
|---|---|
| Try to find the object before going to the kitchen table | Success |

**Command:** *Robot please retrieve the tropical juice from the side table, grasp the apple from the end table, and speak*

| Minimal | Tuned |
|---|---|
| Try to grasp the tropical juice before detecting
Try to grasp the apple before releasing tropical juice
Ambiguous sentence ("*Activate speech function.*") | Success |

## A.3 Results in RoboCup@Home

Table 4 shows the competition results with our system in RoboCup@Home Japan Open (RCJ) 2023 and RoboCup@Home (RC) 2023. See section 3.3 for detailed explanations.

Table 4: RoboCup@Home DSPL Results of our team. In our trial in RCJ 2023, the team scored 170 points, the perfect score for the second to the most challenging category (Category 2). This led the team to win the first prize both in GPSR task and overall in RCJ 2023.

| | GPSR | Overall |
|---|---|---|
| RCJ 2023 | 1st | 1st |
| RC 2023 | 2nd | 3rd |

## A.4  Experimented Commands in section 4.3

Table 5 is a list of commands used in experiments described in section 4.3. The checkmark (✓) in the table indicates that the command and its setups have characteristics of the corresponding failure modes in section 4.1.

Table 5: Commands tested in section 4.3. Blue text indicates the information to navigate is sufficient, and red text indicates the information to navigate is insufficient. Our self-recovery prompting pipeline successfully recovered from all failure cases.

| | Command | Failure Modes | | |
|---|---|---|---|---|
| | | M1 | M2 | M3 |
| 1 | Could you bring me an apple from the side table? | ✓ | | ✓ |
| 2 | Hi HSR, I am starting to feel hungry so could you grab an apple from dining table and put it on my desk? I will be there in a moment. | ✓ | | ✓ |
| 3 | I lost my mug so could you find it for me? | ✓ | | |
| 4 | Thank you, HSR. I am getting tired. Could you prepare a fruit for me on the side table? I will have some rest at the sofa in a moment. | ✓ | | ✓ |
| 5 | Could you help me find the apple that I bought the other day? Ashley might know where it is, so maybe you can ask her if she knows where it is. When you find it, please bring it to me. | ✓ | | |
| 6 | Could you bring me the apple from the stair-like shelf? | | ✓ | |
| 7 | Could you look for Ashley in the dining room and ask her if she wants dinner at home tonight? | | | ✓ |

