# OpenReview forum: "Self-Recovery Prompting: Promptable General Purpose Service Robot System with Foundation Models and Self-Recovery"
_robot-learning.org/CoRL/2023/Workshop/TGR — CoRL 2023 Workshop TGR Poster_

### Official Review · Reviewer_omcH · 2023-10-16

**Rating:** 8
**Confidence:** 3

**Review:**

This paper propose a system for service robots with self-recovery mechanism, and the provided real-world experiments demonstrate its effectiveness. Self-recovery is an important topic for real world robot applications. It would especially necessary for generalist robots that want to handle complex situations.

---

### Decision · Program_Chairs · 2023-10-20

**Decision:**

Accept (Poster)

**Comment:**

Great paper and closely aligned topic!